# Stiff Person Syndrome and Gluten Sensitivity

**DOI:** 10.3390/nu13041373

**Published:** 2021-04-20

**Authors:** Marios Hadjivassiliou, Panagiotis Zis, David S. Sanders, Nigel Hoggard, Ptolemaios G. Sarrigiannis

**Affiliations:** 1Academic Department of Neurosciences, Sheffield Teaching Hospitals NHS Trust, Royal Hallamshire Hospital, Glossop Road, Sheffield S10 2JF, UK; zis.panagiotis@ucy.ac.cy (P.Z.); p.sarrigiannis@sheffield.ac.uk (P.G.S.); 2Academic Department of Gastroenterology, Sheffield Teaching Hospitals NHS Trust, Royal Hallamshire Hospital, Glossop Road, Sheffield S10 2JF, UK; david.sanders1@nhs.net; 3Department of Infection, Immunity & Cardiovascular Disease, University of Sheffield, Sheffield S10 2JF, UK; n.hoggard@sheffield.ac.uk

**Keywords:** stiff person syndrome, anti-GAD antibodies, gluten sensitivity, coeliac disease, cerebellar ataxia, gluten free diet

## Abstract

Stiff person syndrome (SPS) is a rare autoimmune disease characterised by axial stiffness and episodic painful spasms. It is associated with additional autoimmune diseases and cerebellar ataxia. Most patients with SPS have high levels of glutamic acid decarboxylase (GAD) antibodies. The aetiology of SPS remains unclear but autoimmunity is thought to play a major part. We have previously demonstrated overlap between anti-GAD ataxia and gluten sensitivity. We have also demonstrated the beneficial effect of a gluten-free diet (GFD) in patients with anti-GAD ataxia. Here, we describe our experience in the management of 20 patients with SPS. The mean age at symptom onset was 52 years. Additional autoimmune diseases were seen in 15/20. Nineteen of the 20 patients had serological evidence of gluten sensitivity and 6 had coeliac disease. Fourteen of the 15 patients who had brain imaging had evidence of cerebellar involvement. Twelve patients improved on GFD and in seven GFD alone was the only treatment required long term. Twelve patients had immunosuppression but only three remained on such medication. Gluten sensitivity plays an important part in the pathogenesis of SPS and GFD is an effective therapeutic intervention.

## 1. Introduction

Glutamic acid decarboxylase (GAD) is the enzyme involved in the synthesis of the inhibitory neurotransmitter gamma-aminobutyric acid (GABA). GAD is found in both the central and peripheral nervous systems and in pancreatic beta cells [1]. GAD antibodies were first detected and characterised in children with newly diagnosed insulin dependent diabetes mellitus (IDDM) [2]. These were shown to be reacting with pancreatic islet cell proteins.

The first neurological disease to be associated with anti-GAD antibodies was stiff-person syndrome (SPS) [3]. SPS is a very rare autoimmune neurological disease, clinically characterised by axial rigidity, often resulting in hyperlordosis, painful spasms and anxiety. It belongs to a spectrum of CNS hyperexcitability syndromes. SPS is often associated with additional autoimmune diseases such as hypothyroidism, IDDM, pernicious anaemia and others. The majority of patients with SPS have anti-GAD antibodies. Anti-GAD antibodies have also been found in some cases of sporadic idiopathic ataxias [4]. Their presence implies an autoimmune pathogenesis raising the possibility of therapeutic interventions with immunosuppressive medication.

We have previously made a connection between anti-GAD associated diseases and gluten sensitivity (GS) including coeliac disease (CD) [5]. We were also able to show considerable overlap between anti-GAD ataxia and gluten ataxia (70% of patients with anti-GAD ataxia are gluten sensitive), and we have demonstrated that gluten free diet (GFD) is an effective therapeutic intervention in such patients [6]. In this report we share our experience in managing and treating patients with SPS and in particular highlighting the overlap between SPS, gluten sensitivity and CD as well as reporting the therapeutic effect of gluten free diet (GFD).

## 2. Methods

This report is based on a retrospective observational case series of patients regularly attending our specialist clinics (GS/neurology, neuroimmunology and ataxia). The South Yorkshire Research Ethics Committee has confirmed that no ethical approval is indicated given that all investigations/interventions were clinically indicated and did not form part of a research study. All patients were identified from these clinics by one of the authors (MH) who is in charge of the clinical care of all these patients. The patients have been looked after for over 25 years and are under regular follow up by the same consultant neurologist. The diagnosis of SPS was based on the typical clinical features (stiffness, axial rigidity, episodic painful spasms) in addition to the presence of high titre of anti-GAD antibodies (>2000 U/mL) and neurophysiological evidence of CNS hyperexcitability (continuous motor unit activity on EMG and/or abnormal blink reflex).

Serological testing in addition to anti-GAD antibodies, included antigliadin antibodies (AGA, Phadia), TG2 (Phadia), endomysium antibodies (EMA, Werfen) and TG6 antibodies (Zedira). Those patients with one or more positive antibodies were offered gastroscopy and duodenal biopsy to establish the presence of enteropathy (triad of villous atrophy, crypt hyperplasia, increased intraepithelial lymphocytes). All patients with positive serology for gluten sensitivity were advised to adopt a GFD irrespective of the presence of enteropathy. They were all reviewed by an experienced dietitian and given detail advice on GFD. Depending on clinical response after GFD some patients were also offered treatment with immunosuppression. This included intravenous immunoglobulins, azathioprine, mycophenolate, rituximab, plasma exchange and cyclophosphamide.

All patients underwent brain imaging with MRI, some also had MR spectroscopy of the cerebellum. Cerebellar involvement in the context of SPS is almost universal but often under-reported by patients because their most disabling symptoms are those of stiffness and painful spasms.

## 3. Results

We identified 20 patients with SPS over the last 25 years. There were 11 female and 9 male patients. Mean age at onset of symptoms was 52 (range 37–69 years). The presenting symptoms included primarily leg stiffness in 12, truncal stiffness in 5, painful spasms in 2 (painful spasms became a prominent feature in most patients later on in the disease), ataxia in 2 (later a feature in 17 patients) and one leg stiffness in one. Additional autoimmune diseases (apart from GS and CD) were present in 15 patients with 11 having hypothyroidism, 7 having IDDM, 2 having myasthenia gravis, 2 having Sjogren’s syndrome, 1 pernicious anaemia and 1 psoriatic arthropathy (some patients had more than one autoimmune disease). Serological evidence of GS (one or more of AGA, EMA, TG2 and TG6 antibodies) were found in 19 of the 20 patients (95%). The diagnosis of gluten sensitivity was made in Sheffield by the authors and only one of the patients had a pre-existing diagnosis of CD. Fourteen patients underwent gastroscopy and duodenal biopsy. Six patients had evidence of CD on biopsy, the remaining 8 had a normal mucosa. Only 3 of the 19 patients with GS/CD had any gastrointestinal symptoms (diarrhoea, bloating, abdominal pain) attributed to GS/CD. All of these 3 had coeliac disease on biopsy and in all 3 the gastrointestinal symptoms improved on a GFD. The above results are summarised in Table 1.

Neurophysiology showed continuous motor unit activity in keeping with SPS in 13 of the patients. Three patients did not have neurophysiology but the clinical phenotype in combination of high anti-GAD was sufficient to enable the diagnosis. The remaining 4 patients had normal neurophysiology, but this was done after the patients were established on antispasmodic medication. Abnormal blink reflex was seen in 2 patients, but this was only performed in 4 patients.

GFD was recommended to all 19 patients with gluten sensitivity. In 12/19 patients (5 with CD and 7 with GS and no enteropathy) the GFD was found to be beneficial to their SPS and ataxia symptoms (reduced frequency of spasms, stabilisation of mobility, rigidity and improved ataxia). Five patients did not derive any benefit. Two patients did not go on GFD. In 7 patients (3 with CD and 4 with GS and no enteropathy) GFD alone was the only treatment needed to keep their symptoms under control and stabilise their condition. Immunosuppression was used in 12 patients. This included intravenous immunoglobulins (IVIgs) in 9, mycophenolate in 5, plasma exchange in 2, azathioprine in 1, rituximab in 1 and cyclophosphamide in 1 (some patients had more than one immunosuppressive medication). One patient underwent autologous stem cell transplantation because nothing else was effective. This resulted in stabilisation [7]. Only one of the 9 patients that received immunoglobulins found it beneficial long term. In the remaining, oral immunosuppression was felt to be more effective than repeat IVIg’s. All apart from 3 patients were taking antispasmodic medication. Six on 3 different antispasmodics, 5 on 2 and 6 on one. The most commonly used antispasmodics were baclofen (14), dantrolene (9), diazepam (8) and tizanidine (1). One patient with very resistant disease also had botox injections and found entonox very helpful for the painful spasms. Ten patients were on Gabapentin. The addition of this medication seemed to offer some additional benefit in partially controlling/stabilising their symptoms.

MR spectroscopy of the cerebellum was done in 15 patients. Evidence of reduced NAA/Cr ratio in the vermis and/or hemisphere was seen in 14 (mean NAA/Cr from the vermis was 0.9 (range 0.79–1.12) and from the hemisphere 0.95 (range 0.74 to 1.1), normal ratio should be above 1, in keeping with cerebellar dysfunction. Only one of the 15 patients had normal spectroscopy (NAA/Cr vermis 1.03 and hemisphere 1.23). Follow up MR spectroscopy was available in 10 patients who were also GS (5) or had CD (5), after staring GFD (no immunosuppression). Eight showed improved MR spectroscopy (NAA/Cr are ratio) from the vermis whilst 2 showed deterioration.

At the time of writing this report 5 patients had died. Three as a result of SPS (2 in hospital with complications of immobility, one had respiratory arrest resulting in brain hypoxia during a severe chest spasm causing respiratory arrest). One died of myocardial infarction (also had IDDM) and the other due to COVID pneumonia. Of the remaining 15 patients 11 remain mobile with minimal walking aids (use of one stick) and 4 are wheel-chair bound.

Out of the 15 patients who are still alive, 12 are still on a strict GFD, and only 3 are on immunosuppression (1 regular IVIgs, 2 on mycophenolate).

### Illustrative Clinical Case

This 60-year-old man was referred to our centre 4 years ago with an established diagnosis of SPS. He had received high dose steroids on several occasions after being admitted with painful spasms that rendered him bed bound. The episodes of severe spasms had become so disabling that the patient had become understandably anxious and fearful of leaving the house.

At the time of the initial assessment, he was on no regular immunosuppression, but the referring neurologist was planning the introduction of IVIgs. On clinical examination in addition to severe stiffness in both legs and exaggerated lordosis he had evidence of incoordination with nystagmus on lateral gaze, finger nose and heel to sheen ataxia. He also had gait ataxia. He was using a wheelchair when out of his home.

He had a history of hypothyroidism. Investigations showed high titre of anti-GAD antibodies (>2000 U/mL), with neurophysiological evidence of continuous motor unit activity on EMG which, in combination with the clinical presentation led to the diagnosis of SPS.

In addition to the high levels of anti-GAD antibodies he was positive for AGA, EMA, TG2 and TG6 antibodies. Duodenal biopsy confirmed the presence of CD. MR spectroscopy confirmed abnormal NAA/Cr ratio over the vermis in keeping with the clinical findings of gait ataxia (Figure 1). He started on a strict GFD. Within the first 6 months he observed significant reduction in the painful spasms. This was without any other medication. His mobility improved and he was now able to walk using a frame and also go out of his home. After a year on GFD his gluten sensitivity-related antibodies were no longer present. Repeat MR spectroscopy of the cerebellar vermis showed improved NAA/Cr ratio (from 0.84 to 0.90, normal ratio should be over 1) reflecting the improved mobility. He is still stable (3 years after the introduction of GFD) and not requiring any immunosuppression.

## 4. Discussion

We have previously reported an association between anti-GAD related diseases and GS/CD [5]. We have also recently published our experience in the management of 50 patients with anti-GAD ataxia where we have again shown a significant overlap between anti-GAD ataxia and gluten ataxia (70% of patients with anti-GAD ataxia were gluten sensitive) [6]. Furthermore, we have shown that patients with anti-GAD ataxia who are gluten sensitive respond well to strict GFD, with improvement of the ataxia. In this report we present our experience of managing 20 patients with SPS, primarily to highlight the strong association with gluten GS/CD (95% positive for one or more gluten sensitivity-related antibodies and at least 30% having CD) and demonstrate that GFD has a therapeutic role to play. The prevalence of AGA antibodies in the healthy population was 12% and that of CD 1% [8].

The association between SPS and gluten sensitivity cannot be simply explained on the basis of an association of two autoimmune diseases by chance. Nineteen of these 20 patients (95%) with SPS were found to be gluten sensitive. In addition, we have shown that GFD has an important therapeutic role to play in these patients, suggesting that GS/CD must play a role in the pathogenesis of SPS.

As per the case highlighted in this report, the majority of patients who are on GFD have found this intervention helpful in controlling their SPS symptoms.

In our experience, cerebellar involvement in the context of SPS appears to be universal in these patients. In fact, cerebellar ataxia in isolation is a commoner manifestation of anti-GAD related diseases than SPS based on our experience; the number of patients with anti-GAD ataxia we have treated is 50 as opposed to 20 with SPS. However, there may be some referral bias given that our unit is one of the National Ataxia Centres in the UK.

Cerebellar involvement in the context of SPS may have an important pathophysiological role to play; the output of the cerebellum is all inhibitory. Any dysregulation of such output could potentially result in a state of CNS hyperexcitability. This state of hyperexcitability is particularly prominent in the immune ataxias by contrast to the genetic or degenerative ataxias [9]. It can manifest with rigidity and spasms, as is the case in SPS but also with cortical myoclonus as is often seen in cases of refractory CD [10]. Additional clinical markers of hyperexcitability include brisk reflexes and exaggerated startle. It is possible that the selective involvement of different cerebellar cell populations may explain why the state on brain hyperexcitability is more often seen in immune rather than genetic ataxias.

The association between anti-GAD antibodies and gluten sensitivity merits further consideration. Ventura et al. have made the observation that the prevalence of additional autoimmune diseases in children with CD is significantly lower than in those patients with CD diagnosed in adulthood [11]. They concluded that GFD may reduce the risk of developing additional autoimmune diseases later on in life. This observation echoes our previously observed reduction in anti-GAD antibodies in patients with anti-GAD related diseases and gluten sensitivity who go on a strict GFD [5].

The pathological role of anti-GAD antibodies in the genesis of SPS and ataxia is unclear. Since GAD65 (the GAD isoform implicated in these diseases) is intracellular and is associated with a range of neurological conditions, some have argued that anti-GAD65 antibodies have no pathogenic role to play. On the other hand, recent physiological studies in vitro and in vivo have demonstrated that binding of GAD by anti-GAD antibodies induces loss of GAD functions relating to GABA release, leading to the development of cerebellar ataxia [12]. Given these observations the question remains as to why should GFD be beneficial in those patients with gluten sensitivity and SPS. The response to GFD in those patients who are gluten sensitive suggests that gluten sensitivity may be driving the immune response that results in SPS.

There are no clear-cut evidence-based guidelines for the treatment of SPS. It has been shown that regular IVIgs can be beneficial [13]. This has not been our experience. Only 2 of the 20 patients reported here found regular IVIgs of some sustained benefit. Immunosuppression with mycophenolate has been beneficial for some patients but by far the most effective therapeutic intervention has been the GFD.

In terms of symptom relief most patients require a combination of one or often several antispasmotics of which in our experience dantrolene, baclofen and diazepam offer the best combination. We also have used gabapentin more as a “disease modifier” simply on the theoretical benefit based on its mode of action.

To conclude, 95% of our patients with SPS who have been under regular review at our centre have evidence of gluten sensitivity or CD and have benefited from a strict GFD. The diagnosis of gluten sensitivity relies on a range of antibodies and the use of EMA or TG2 antibodies alone, whilst sufficient to diagnose CD, cannot diagnose gluten sensitivity without enteropathy. This is an important consideration because GFD is beneficial for patients with SPS who are gluten sensitive irrespective of the presence of enteropathy.

## Figures and Tables

**Figure 1 nutrients-13-01373-f001:**
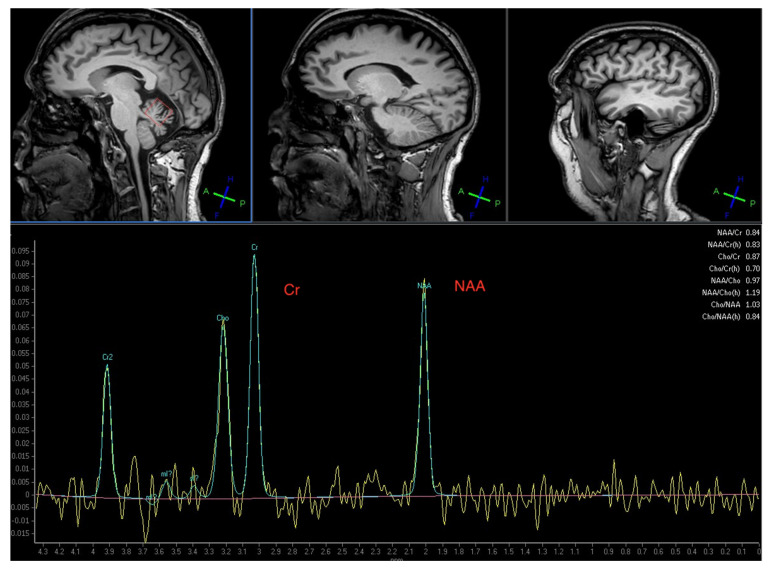
Magnetic Resonance Spectroscopy of the cerebellar vermis from the illustrative clinical case (see text) showing a significant reduction of the N-Acetyl-Aspartate to Creatine ratio (NAA/Cr) at 0.84 (normal should be above 1). All but one of these patients with stiff person syndrome (SPS) had abnormal spectroscopy of the cerebellum highlighting the fact that cerebellar involvement is universal in SPS.

**Table 1 nutrients-13-01373-t001:** Clinical characteristics, investigations and outcomes of 20 patients with stiff person syndrome.

number of patients with SPS reported	20
male: female	9:11
mean age at symptom onset (range)	52 (37–69 years)
additional autoimmune diseases	11 hypothyroidism, 7 IDDM, 2 myasthenia gravis, 2 Sjogren’s, 1 pernicious anaemia, 1 psoriatic arthropathy
serological evidence of gluten sensitivity (patients may have been positive for more than one antibody)	19/20 (95%) (17 AGA, 7 TG6, 5 TG2, 5 EMA)
coeliac disease (out of 14 patients who had duodenal biopsy)	6
abnormal neurophysiology showing continuous motor unit activity	13, normal in 4, not done in 3
abnormal blink reflex	abnormal in 2, only done in 4
abnormal MR spectroscopy of the cerebellum suggestive of cerebellar involvement	14/15 (93%)
Improved on gluten free diet	12/19 (63%)
number that tried immunosuppression (still on immunosuppression)	12 (3)

Please note that not all patients were tested for TG6 antibodies. (IDDM-insulin dependent diabetes mellitus, AGA-antigliadin antibodies, TG6- transglutaminase 6 antibodies, TG2-transglutaminase 2 antibodies, EMA- endomysium antibodies).

## Data Availability

Anonymised data can be provided on request.

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
