# Peer review of "Stiff Person Syndrome and Gluten Sensitivity"

_nutrients, 2021, doi:10.3390/nu13041373_

Round 1
Reviewer 1 Report
This is a comprehensive assessment of an interesting question. My only concern relates to case ascertainment. Given the author's significant contribution to the study of the connections between gluten sensitivity and neurological disease, the source of the patients raises questions. Are these patients selected in some way or are they typical of SPS patients?
Author Response
This is a comprehensive assessment of an interesting question. My only concern relates to case ascertainment. Given the author's significant contribution to the study of the connections between gluten sensitivity and neurological disease, the source of the patients raises questions. Are these patients selected in some way or are they typical of SPS patients?
We thank the reviewer for the very helpful and valid comment. All apart from one of the 20 patients reported here, with typical SPS, were diagnosed with gluten sensitivity and/or coeliac disease after being investigated in Sheffield. Only one of the patients had a pre-existing diagnosis of CD. We have added this information to the results section.
Reviewer 2 Report
This is a very interesting retrospective observational case series that calls attention to a possible link between gluten sensitivity and stiff person syndrome (SPS). The findings may help elucidate the pathogenesis of SPS and offer hope for better treatment options for patients suffering from this extremely rare and debilitating disease. Overall, the manuscript is well written, but it could be improved by additional clarification of its methods and results.
- The authors appear to label patients as gluten sensitive solely on the basis of serologies. In contrast, the Salerno Experts define non-celiac gluten sensitivity (NCGS) as a clinical syndrome with symptoms that respond to withdrawal of gluten from the diet in the absence of celiac disease (CD) or wheat allergy. Did the patients labeled gluten sensitive have other symptoms that were attributed to gluten ingestion/improved after initiation of a gluten-free diet (GFD)?
- Is the presence anti-gliadin antibodies in SPS patients truly significant? The prevalence of these antibodies in the general population and other autoimmune conditions would serve as useful comparators.
- Was there a standard way that improvement in symptoms/benefit after GFD was measured? SPS often follows a relapsing-remitting course, which can make it difficult to assess response to interventions.
- Could you please report how many of the patients with confirmed CD (n = 6) responded to GFD compared to the number of those confirmed not to have CD (n = 8)? Of the 7 patients who responded to GFD and needed no other treatment, how many had CD vs. NCGS?
- I am confused about breakdown of gluten sensitive patients with SPS (starting on line 128). Twelve had at least a partial response to GFD; five had no response to GFD; one did not go on a GFD. That only adds up to 18 patients. What happened to the final patient?
- In the table, the authors write that 12 patients “improved on GFD alone.” I think this is confusing/misleading; symptoms were controlled without additional treatment in only 7 patients (line 132). Similarly, in the discussion, the authors write that “the majority of patients... on GFD have found this intervention alone sufficient to control their SPS symptoms.” Again, I do not see how the data support this claim. GFD may have been associated with improvement in symptoms in the majority of patients, but it was clearly not sufficient in the majority of patients if only 7 required no other treatments.
- In line 275, the authors seem to say that 95% of the SPS patients benefited from GFD. This is not accurate.
Author Response
- The authors appear to label patients as gluten sensitive solely on the basis of serologies. In contrast, the Salerno Experts define non-celiac gluten sensitivity (NCGS) as a clinical syndrome with symptoms that respond to withdrawal of gluten from the diet in the absence of celiac disease (CD) or wheat allergy. Did the patients labeled gluten sensitive have other symptoms that were attributed to gluten ingestion/improved after initiation of a gluten-free diet (GFD)?
We appreciate the reviewer’s point. We deliberately avoided the use of the term NCGS because this term applies to gastroenterology cases (ie patients who have gastrointestinal symptoms that improve on GFD). This is different to neurology cases where the presentation can be purely neurological with minimal if any gastrointestinal symptoms. Furthermore, in the neurology cases, in addition to the neurological dysfunction (eg ataxia or neuropathy), patients have to be positive for at least one gluten sensitivity-related antibody to be labelled as gluten sensitive. This is not true for the gastroenterology NCGS cases where patients may have no serological markers of GS but simply clinically improve on GFD (as per Salerno criteria). We have provided the data of the response to GFD in this cohort but we have now also added the limited data (see results) on improved GI symptoms for those 3 patients with CD who had GI symptoms (the vast majority did not).
- Is the presence anti-gliadin antibodies in SPS patients truly significant? The prevalence of these antibodies in the general population and other autoimmune conditions would serve as useful comparators.
The prevalence of 95% SPS patients having such antibodies is indeed significant when compared to 12% in the healthy population (based on the same assay). The prevalence of CD of at least 30% in this cohort is also significant when compared to 1% in the healthy population. We have added this information to the discussion section and added a reference (reference 8). The problem of comparing such AGA figures with other publications that focus on AGA and other autoimmune diseases is that the assays are different and thus non-comparable.
- Was there a standard way that improvement in symptoms/benefit after GFD was measured? SPS often follows a relapsing-remitting course, which can make it difficult to assess response to interventions.
A very valid point. The fluctuating course of the disease and the frequent adjustments in the use of antispasmodic medication makes it difficult to monitor progress by the use of specific spasticity scales (basic and rather limited) or other symptom outcomes. In some respect the decision to use immunosuppressive medication or not may be one way of judging the effectiveness of the gluten free diet alone, which is why we have included this information. In addition, we do have some data on the MR spectroscopy of the cerebellum before and after GFD showing improvement in 8 out of 10 patients who adopted GFD alone (this includes the illustrative case). We have added the data to the results.
- Could you please report how many of the patients with confirmed CD (n = 6) responded to GFD compared to the number of those confirmed not to have CD (n = 8)? Of the 7 patients who responded to GFD and needed no other treatment, how many had CD vs. NCGS?
We have added all this additional information as suggested
- I am confused about breakdown of gluten sensitive patients with SPS (starting on line 128). Twelve had at least a partial response to GFD; five had no response to GFD; one did not go on a GFD. That only adds up to 18 patients. What happened to the final patient?
Apologies. There were 2 patients who opted not to go on a GFD. We have corrected this.
- In the table, the authors write that 12 patients “improved on GFD alone.” I think this is confusing/misleading; symptoms were controlled without additional treatment in only 7 patients (line 132). Similarly, in the discussion, the authors write that “the majority of patients... on GFD have found this intervention alone sufficient to control their SPS symptoms.” Again, I do not see how the data support this claim. GFD may have been associated with improvement in symptoms in the majority of patients, but it was clearly not sufficient in the majority of patients if only 7 required no other treatments.
We accept the valid comment of the reviewer. We have removed the word “alone” from the table and also changed the wording in the discussion section.
- In line 275, the authors seem to say that 95% of the SPS patients benefited from GFD. This is not accurate.
We have altered the wording accordingly.